# Not All Bits have Equal Value:
# Heterogeneous Precisions via Trainable Noise

**Pedro Savarese**
TTI-Chicago
savarese@ttic.edu

**Xin Yuan**
University of Chicago
yuanx@uchicago.edu

**Yanjing Li**
University of Chicago
yanjingl@uchicago.edu

**Michael Maire**
University of Chicago
mmaire@uchicago.edu

## Abstract

We study the problem of training deep networks while quantizing parameters and activations into low-precision numeric representations, a setting central to reducing energy consumption and inference time of deployed models. We propose a method that learns different precisions, as measured by bits in numeric representations, for different weights in a neural network, yielding a heterogeneous allocation of bits across parameters. Learning precisions occurs alongside learning weight values, using a strategy derived from a novel framework wherein the intractability of optimizing discrete precisions is approximated by training per-parameter noise magnitudes. We broaden this framework to also encompass learning precisions for hidden state activations, simultaneously with weight precisions and values. Our approach exposes the objective of constructing a low-precision inference-efficient model to the entirety of the training process. Experiments show that it finds highly heterogeneous precision assignments for CNNs trained on CIFAR and ImageNet, improving upon previous state-of-the-art quantization methods. Our improvements extend to the challenging scenario of learning reduced-precision GANs.

## 1 Introduction

Scaling up the size of deep neural networks offers returns to accuracy in tasks across a range of application areas, including computer vision and natural language processing. Expanding network depth from tens of layers [23, 44] to a hundred [13, 52] revolutionized image classification and object detection [9, 14]. Following the 340M parameter BERT model [4], the size of state-of-the-art language models has increased by multiple orders of magnitude: GPT-2 with 1.5B [40], GPT-3 with 175B, and recently Megatron-Turing with 530B parameters [45] – more than a 1000-fold increase in size over BERT. Larger models increase the computational cost of inference, posing challenges to their deployment in real-world applications.

To ameliorate these costs, multiple research avenues have explored methods for constructing more compact deep networks while preserving accuracy comparable to larger models. These include, *e.g.,* architectural building blocks with sparser connectivity [17, 37, 54, 16] or efficient dimensionality reduction stages [20]. Another category of approaches shrinks larger models via pruning [11, 31, 29, 8, 24, 43], reducing precision [12, 19, 51], or binarization [18, 41, 35, 38]. Parallel efforts on neural architecture search [57, 36, 27, 32, 28, 49] optimize coarser aspects of model structure by automating exploration of hyperparameters governing network size and configuration of layers.

36th Conference on Neural Information Processing Systems (NeurIPS 2022).

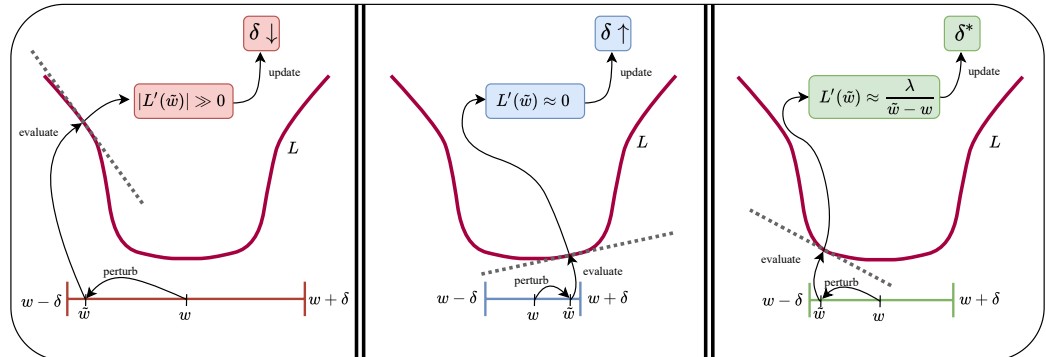

Figure 1: **Weight noise as a differentiable proxy for precision.** A learnable magnitude $\delta$ scales uniform random noise added to weight $w$ during training. The width of the basin over which it is possible to perturb $w$ without increasing task loss $L$ drives learning of $\delta$. After training, we reduce the bit precision of the numeric representation of $w$ as much as possible, with the constraint of remaining in the $(w - \delta, w + \delta)$ range. ***Left:*** A random perturbation $\tilde{w}$ increases loss, driving a decrease in $\delta$. ***Middle:*** Perturbation leaves loss unchanged, driving an increase in $\delta$. ***Right:*** Noise level $\delta$, and the corresponding implied precision, stabilize when matched to the size of the basin.

Yet, these methods fall short of providing a complete framework for producing efficient networks. Pruning, quantization, and binarization are often implemented as heuristic post-processing transformations that operate on a trained network [12, 25] or require special training procedures [18, 41, 53, 56]. Section 2 provides a broader survey. The lack of modular approaches to optimizing the network structure at this finest level of detail complicates potential combination with neural architecture search techniques for selecting the functional form of layers and building blocks at coarser granularity.

We introduce a principled framework for training compact neural networks, which encompasses pruning, quantization, and binarization. Each is a special case of learning a more general heterogeneous mixed-precision architecture, wherein a set of auxiliary parameters govern the bit precision of each weight in the original network architecture. Our approach learns auxiliary parameters simultaneously with learning the original parameters (weights), with training driven entirely by stochastic gradient descent. In addition to modularity, this strategy provides two crucial benefits:

- Unlike schemes which post-process a trained network, our method makes the entirety of the training process aware of the goal of producing a reduced-precision model. Learning weights in the same context as quantization avoids divergence between training and inference scenarios.

- The auxiliary parameters can freely participate in the definition of the loss being optimized during training. We can craft loss terms that approximate actual costs (*e.g.,* in computation or energy usage) of inference as determined by the current configuration of numeric precisions. Combined with the task-specific loss term, we can choose how to trade-off accuracy and efficiency.

Key to our technical approach is correspondence between a notion of noisy weights and numeric precision. Specifically, during training, we instantiate each network weight as a full-precision value plus uniform random noise of a learnable magnitude. These noise magnitudes are precisely the auxiliary parameters. The more noise that can be injected into the value of a particular weight without increasing loss, the lower the precision required to represent that weight.

Figure 1 illustrates the intuition behind this correspondence. During training, noise magnitudes serve as a differentiable proxy for discrete bit precisions. Upon completion of training, the learned noise magnitudes dictate replacement of noisy weights with fixed, reduced-precision values. Though related to Bayesian neural networks, our formulation differs substantially from prior work on Bayesian approaches to network pruning and quantization [30]. Section 3 provides details, with complete derivations in Appendix A.

Our method is able to allocate precisions under any level of granularity (*e.g.,* per-parameter, per-layer) by sharing the trainable noise magnitude variables across different parameters. Although a per-layer precision scheme is a more natural fit for current GPUs, new hardware designs can leverage more

fine-grained and heterogeneous schemes [34, 46, 21]; hence, we focus on the limit of granularity where each parameter has its own precision. This setting has the potential to achieve significant energy efficiency and run-time advantages over the per-layer setting without sacrificing accuracy, as suggested by our empirical evidence. Future commercially available accelerators, based on the latest hardware research, could make these advantages real.

Results in Section 4 show that, on standard benchmarks such as creating weight-quantized CNNs for image classification on CIFAR-10 and ImageNet, our approach outperforms competitors by achieving higher accuracy using fewer bits per parameter. Further experiments show similar improvements when training GANs. We also show empirically that, when extended to quantize hidden activations of a network, our method is able to find heterogeneous precision schemes and outperforms well-established activation quantization methods.

The source of our advantages may span both the ability to simultaneously account for precision and task objectives within the training dynamics, as well as the ability to learn heterogeneous weight precisions at the finest level of granularity.

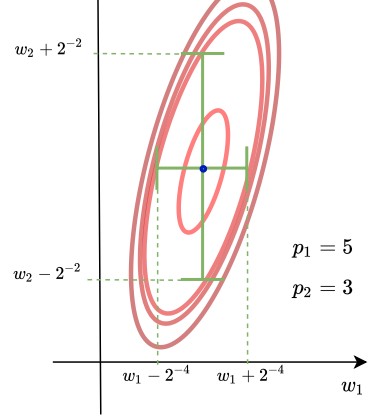

## 2 Related Work

Most approaches in the quantization literature assume that a predefined precision is given prior to training [55, 53, 10, 51, 19, 33, 3, 26, 12, 6], designating the number of bits used to represent each parameter of the network. The focus of these works is typically to circumvent the obstacle posed by the non-differentiability of the quantization function, as this makes first-order methods inapt to train quantized models due to the lack of meaningful gradients.

Figure 2: Multidimensional example where different precisions should be assigned to each parameter due to their distinct perturbation limits. Note the connection between each parameter's perturbation limit, the width of the level curve in the parameter's axis, and the allocated precision for each parameter.

DoReFa [55] updates real-valued weights using gradients w.r.t. their quantized forms (*i.e.,* a straight-through estimator [1]), enabling training over quantized parameter values with gradient descent. LQ-Nets [53], PACT [3], and LSQ [6] introduce further flexibility to quantization by also optimizing the quantization step size, which we refer by *quantization scale* throughout our work, *i.e.,* the real value that each bit corresponds to is also learned.

DSQ [10] proposes a smooth approximation to the quantization mapping, which allows weights to be directly trained with SGD without straight-through estimators. SLB [51] introduces per-parameter auxiliary variables that induce distributions over values that quantized weights can take – these variables are trained with gradient descent and a final quantized model is created by approximating each weight's distribution by a point-mass.

More recently, some works on quantization have explored ways to assign different precisions to parameter groups in a neural network [48, 47, 5, 50] – such methods, however, remain a minority in the quantization literature. DNAS [48] uses neural architecture search to map candidate precisions to different parameter groups. HAQ [47] uses reinforcement learning to learn a quantization policy. HAWQ [5] uses second-order information to allocate precisions to different parameter groups.

Closest to our work is BSQ [50], which introduces new variables that are trained with gradient-based methods, which are then used to allocate precisions throughout the network. It operates by first mapping each real-valued weight to a bitstring variable whose length is chosen a-priori. These variables are then iteratively trained with gradient descent and pruned based on their magnitudes, resulting in a decrease in precision whenever a component (a bit) of the bitstring is removed. Finally, the bitstrings are mapped back to weights, and the quantized model undergoes fine-tuning by adopting straight-through estimators for the quantized weights.

Lastly, Fan et al. [7] improve the performance of quantized networks by randomly selecting a subset of weights to be quantized during training while keeping the remaining parameters in their original

---
**Algorithm 1** SMOL
---
1: **procedure** SMOL ($w \in \mathbb{R}^d, L : \mathbb{R}^d \rightarrow \mathbb{R}, \lambda \in \mathbb{R}, p_{init} \in \mathbb{N}$)
2:     Instantiate parameters $s \in \mathbb{R}^d$ and initialize $s_{init} = -\ln\left(2^{p_{init}-1} - 1\right)$
3:     **for** $t = 1 \rightarrow T_1$ **do**
4:         Sample $\epsilon^{(t)} \sim \mathcal{U}^d(\pm 1)$
5:         Compute $\mathcal{L}(w, s) = L(w + \sigma(s) \odot \epsilon^{(t)}) + \lambda \|\log_2(1 + e^{-s})\|_1$ and $\nabla_{w,s}\mathcal{L}(w, s)$
6:         Update $w$ and $s$ using $\nabla_w \mathcal{L}(w, s)$ and $\nabla_s \mathcal{L}(w, s)$
7:         Clip $w$ to $\pm(2 - \sigma(s))$
8:     **end for**
9:     Set $p = 1 + \text{round}(\log_2(1 + e^{-s}))$
10:     ZPA (Optional): If $|w| < |w - q(w, p)|$ then set $p = 0$ (element-wise)
11:     **for** $t = T_1 \rightarrow T_2$ **do**
12:         Compute $L(w_q)$ and $\nabla_{w_q} L(w_q)$, where $w_q = q(w, p)$
13:         Update $w$ using $\nabla_{w_q} L(w_q)$ instead of $\nabla_w L(w_q)$
14:     **end for**
15: **end procedure**
---

format. While our approach also uses randomness to improve quantization in some fashion, the form of noise samples, along with how and why they are used, are vastly different.

## 3 Method

### 3.1 Allocating Precisions by Optimizing Noise Magnitudes

We consider the problem of allocating a precision $p$ (a positive integer which represents the number of bits) to each weight $w$ of a network under a total precision budget (the total number of bits that can be used to represent the model). If we let $q(w, p)$ denote the quantization of $w$ with $p$ many bits, then our goal is to minimize $L(q(w, p))$ subject to $p \leq C$, where $C$ is our total budget measured in bits and $L$ captures the loss incurred by the weights – typically the cross-entropy loss over a training set.

Although our method is the result of a rigorous sequence of approximations to the objective described above, it admits an intuitive description and motivation that relies on a connection between quantization and perturbations. More specifically, if we assume that a weight $w$ can be perturbed in any direction by at least $\epsilon$ without degrading the performance of the network, then we can safely represent $w$ with $p$ bits as long as the quantization error does not exceed $\epsilon$ – that is, $|w - q(w, p)| \leq \epsilon$.

In other words, if we know the largest perturbation that each weight admits (its *perturbation limit*) without yielding performance degradation, then we can easily assign a precision to each weight: it suffices to choose the smallest number of bits such that the quantization error falls below the perturbation limit of the corresponding weight.

Our method works by estimating the perturbation limit of each parameter to be quantized, which is achieved by directly optimizing the magnitude of the weight perturbations through a novel loss function. We call our method SMOL : **S**earching for **M**ixed-Precisions by **O**ptimizing **L**imits for Perturbations.

Once the perturbation limit of each weight has been estimated through optimization by our method, we assign per-weight precisions by mapping each perturbation limit to a number of bits.

More specifically, for a model with weight parameters $w$ and training loss $L$, our method first introduces new parameters $s$ of the same shape as $w$, and at each iteration $t$ aims to update $w, s$ to decrease an augmented loss function:

$$L\left(w + \sigma(s) \odot \epsilon^{(t)}\right) + \lambda \left\|\log_2(1 + e^{-s})\right\|_1, \tag{1}$$

where $\epsilon^{(t)}$ has the same size as $s$ and whose components are independently sampled from an uniform distribution over the interval $[-1, +1]$, and $\lambda$ controls the trade-off between performance and representational cost – a larger value for $\lambda$ will place more importance on using less bits to represent weights compared to achieving small loss. Note that the noise samples $\epsilon^{(t)}$ are independent across

different iterations, *i.e.,* we draw a new sample $\epsilon^{(t)}$ at each iteration $t$. Here, the term $\sigma(s)$ represents the perturbation limit estimates of the weights $w$: the distance between $w$ and $w + \sigma(s) \odot \epsilon^{(k)}$ is at most $\sigma(s)$ since $\epsilon^k$ is between $-1$ and $+1$.

A key aspect of our proposed loss function lies in the fact that it is fully differentiable w.r.t. to $s$; hence, gradient-based methods like SGD and Adam can be applied off-the-shelf to optimize both the original weights $w$ and the auxiliary variables $s$. This enables the parameters $s$ to be seen as being part of the model itself, allowing for our method to be easily combined with higher-order optimizers, neural architecture search, and other algorithms that operate on networks. After training, we map the final values of $s$ to precisions via some function $\lceil \cdot \rfloor$ that outputs integers *e.g.,* rounding or truncation:

$$p = 1 + \lceil \log_2(1 + e^{-s}) \rfloor. \tag{2}$$

The resulting tensor $p$ will have the same shape as $w$, and its components represent the number of bits assigned to each parameter in $w$. Having allocated a precision to each weight parameter, we can discard the auxiliary variables $s$ and use quantized weights $w_q = q(w, p)$ to compute the model's predictions instead of applying perturbations. This will typically lead to non-negligible changes in the model's activations, which we circumvent by fine-tuning the model by further optimizing $w$ to minimize the training loss. Following prior work, we use straight-through estimators to optimize $w$ since $q$ is non-differentiable, *i.e.,* we set $\nabla_w L = \nabla_{w_q} L$ and perform gradient descent on $w$.

Lastly, note that inverting the $s \rightarrow p$ mapping offers a way to initialize $s$ given a desired initial precision $p_{init}$, more specifically we set $s_{init} = -\ln \left( 2^{p_{init}-1} - 1 \right)$. For example, adopting an initial precision $p_{init} = 8$ results in $s_{init} = -\ln(2^{8-1} - 1) = -\ln(127)$, and the perturbation limit estimate will start as $\sigma(s_{init}) = 2^{1-p_{init}} = 2^{-7}$ for all weights $w$ of the network.

Our method is also applicable if groups of parameters must share the same precision value *e.g.,* the layer-wise setting where all parameters of each layer are represented with the same number of bits. In this case, we assign each parameter group $i$ to a single component $s_i$ of $s$, and when applying perturbations to the weights in the group we scale all the noise samples by the same scalar $\sigma(s_i)$. At the end of training, we map $s_i$ to a single integer $p_i$ via (2) which is shared across the group.

## 3.2 Zero Precision Allocation

A fundamental limitation of having each $i$'th bit map to either $+2^{1-i}$ or $-2^{1-i}$ is that zero weights cannot be represented: the possible values that a 1-bit weight can assume are $\{-1, +1\}$, for a 2-bit weight $\{-1.5, -0.5, +0.5, +1.5\}$, and moreover a quantized weight cannot assume the value $0$ regardless of its precision. However, in our setting – where we can assign a different precision to each weight – we can directly re-define the quantization function $q$ to map any $w$ to $0$ whenever $p = 0$, hence introducing the notion of *zero-precision weights*. This is analogous to the procedure of assigning zero precision to a filter in BSQ, which leads to the corresponding convolution to be completely skipped when computing a model's outputs.

We propose to assign zero precision to a weight $w$ whenever $|w| \leq |w - q(w, p)|$. Since the quantization function $q$ maps $w$ to the closest value that is representable with $p$ bits, whenever a weight's precision is set to zero due to our proposed strategy, the induced quantization error is guaranteed not to increase. This follows since prior to changing $p$ the quantization error is $|w - q(w, p)|$; but once we set $p = 0$, it becomes $|w - q(w, 0)| = |w - 0| = |w|$, which cannot lead to an increase since the condition is precisely $|w| \leq |w - q(w, p)|$.

For example, $q(w = 0.2, p = 2) = 0.5$ since it is the value in $\{-1.5, -0.5, +0.5, +1.5\}$ that is closest to $w = 0.2$, while $q(w = 0.2, p = 0) = 0$ following our re-definition of $q$ yields a quantization error of $|0.2 - 0| = 0.2$. On the other hand, for $p = 2$ we have $|0.5 - 0.2| = 0.3$. Hence, assigning zero precision in this case not only frees up 2 bits but also decreases the quantization error by $0.1$.

One advantage of adopting this procedure to assign zero precisions to weights is that it only changes $q$ and hence the quantized network, therefore not affecting the procedure described in the previous section. In other words, one can allocate precisions to weights by optimizing perturbations as described previously, and then quantize the model separately with and without zero precision allocation.

This results in two quantized models with different precision assignments, hence we can obtain two networks by training the auxiliary variables $s$ only once. The notion of zero precisions also unifies quantization and pruning, since in practice assigning zero precision is equivalent to pruning a weight.

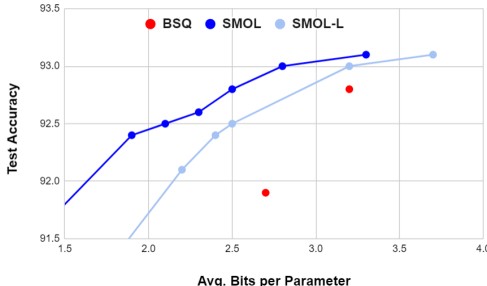
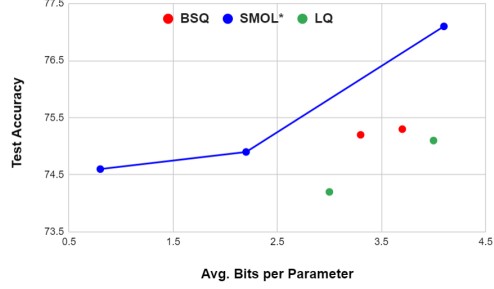

Figure 3: Performance of SMOL and BSQ when quantizing a ResNet-20 trained on CIFAR-10. SMOL-L denotes SMOL with layer-wise precisions.

Figure 4: Performance of SMOL , BSQ, and LQ-Nets when quantizing a ResNet-50 trained on ImageNet. SMOL* denotes SMOL with zero-precision allocation.

# 4 Results

We evaluate our method in the tasks of quantizing CNNs trained for image classification and generation. For all experiments, we adopt an initial precision $p_{init} = 8$, which corresponds to initializing each component of the new parameter $s$ as $-\ln(2^7 - 1) \approx -4.84$.

Table 1: Performance of different quantization methods on ResNet-20 when trained on CIFAR-10. * denotes results with Zero Precision Allocation.

| Method | Precision Granularity | ResNet-20 Avg. Bpp ↓ (Ratio ↑) | ResNet-20 Test Acc. (%) |
|---|---|---|---|
| LQ-FP | | 32.0 (1.0) | 92.1 |
| BSQ-FP | | 32.0 (1.0) | 92.6 |
| LQ-Nets | Network | 1.0 (32.0) | 90.1 |
| DSQ | Network | 1.0 (32.0) | 90.2 |
| SLB | Network | 1.0 (32.0) | 90.6 |
| LQ-Nets | Network | 2.0 (16.0) | 91.8 |
| SLB | Network | 2.0 (16.0) | 92.0 |
| SMOL* | Parameter | 1.3 (24.6) | 91.5 |
| SMOL* | Parameter | 1.7 (18.8) | 92.6 |
| LQ-Nets | Network | 3.0 (10.7) | 92.0 |
| BSQ | Layer | 2.7 (11.9) | 91.9 |
| SMOL-L | Layer | 2.4 (13.3) | 92.4 |
| SMOL | Parameter | 2.1 (15.2) | 92.5 |
| SMOL | Parameter | 2.5 (12.8) | 92.8 |
| BSQ | Layer | 3.2 (10.0) | 92.8 |
| SMOL-L | Layer | 3.2 (10.0) | 93.0 |
| SMOL | Parameter | 2.8 (11.4) | 93.0 |

We use the floor operation to map real-valued precisions to integers: based on our preliminary experiments this more aggressive rounding operation yields more compact models and rarely results in performance degradation.

For all experiments we train the auxiliary parameters $s$ with Adam [22], using the default learning rate of $10^{-3}$ and no weight decay – all its other hyperparameters are set to their default values.

We run SMOL with and without Zero Precision Allocation: allowing for zero precisions results in more compact models, and we use SMOL* to denote results with zero precision allocation. When allocating per-layer precisions, we refer to our method as SMOL-L.

To compare different methods we measure the performance and the size of the resulting network. We report the average number of bits assigned to the model's weights (the sum of all precisions divided by the number of weights), which we refer to as 'average bpp', along with the compression ratio relative to a full-precision model, which equals to 32 divided by the average bpp. Appendix D.4 provides preliminary experiments with Transformers.

## 4.1 Image Classification on CIFAR-10

We first compare SMOL against different quantization methods on the small-scale CIFAR-10 dataset, adopting different networks to evaluate how aggressively each method can quantize weights without degrading the model's generalization performance. Since prior works typically rely on other methods to quantize activations (*e.g.,* using PACT [3] for activations while applying a new method to the weights), we only consider results with full-precision activations for CIFAR-10 as this isolates each method's performance to its ability to quantize weights, leading to a more direct comparison.

Table 2: Performance of SMOL and BSQ on MobileNetV2 and ShuffleNet when trained on CIFAR-10.

| Method | MobileNetV2 | | ShuffleNet | |
|---|---|---|---|---|
| | Avg. Bpp ↓ | Test Acc. | Avg. Bpp ↓ | Test Acc. |
| FP | 32.0 | 94.4 | 32.0 | 90.7 |
| BSQ | 2.8 | 94.1 | 3.4 | 91.8 |
| SMOL | 1.5 | 94.5 | 1.8 | 91.6 |
| SMOL | 1.7 | 94.8 | 2.9 | 92.0 |

We adopt the standard data augmentation procedure of applying random translations and horizontal flips to training images, and train each network for a total of 650 epochs: the precisions are trained with SMOL for the first 350 while the remaining 300 are used to fine-tune the weights while the precisions remain fixed. Note that this training budget assigned to our method is considerably smaller than BSQ's 1000 total epochs which are split between pre-training, precision allocation, and fine-tuning.

Following prior work, we keep the batch normalization parameters in full-precision as they represent a small fraction of the network's total parameters. Like in BSQ, weights of parameterized shortcut connections are also kept in full-precision – namely, shortcuts that consist of $1 \times 1$ convolutions followed by normalization in ResNet-20 and MobileNetV2.

To train the weights we use SGD with a momentum of $0.9$ and an initial learning rate of $0.1$, which is decayed at epochs 250, 500, and 600. We use a batch size of 128 and a weight decay of $10^{-4}$ for ResNet-20, $4 \cdot 10^{-5}$ for MobileNetV2, and $5 \cdot 10^{-4}$ for ShuffleNet.

ResNet-20 results are given in Table 1: SMOL comfortably outperforms BSQ and other competing methods by offering higher performance at lower precision. Comparing against BSQ's $91.9\%$ accuracy at 2.7 bpp, SMOL provides $0.6\%$ higher accuracy at 0.6 lower bpp ($92.5\%$ at 2.1 bpp). With zero-precision allocation (SMOL*), our method outperforms BSQ by $0.7\%$ at 1.0 lower bpp ($92.6\%$ at 1.7 bpp), and matches the performance of the full-precision model with a $18.8\times$ compression ratio.

When allocating layer-wise precisions (SMOL-L), we observe a $0.5\%$ higher accuracy at 0.3 lower bpp compared to BSQ, showing that although per-parameter precisions improve efficiency, our method outperforms the state-of-the-art even when constrained to the less flexible, layer-wise setting (more details in Appendix D.1). Figure 3 shows efficiency curves for BSQ and SMOL-L.

Table 2 presents results for MobileNetV2 and ShuffleNet: SMOL also comfortably outperforms BSQ, offering $0.7\%$ higher accuracy at 1.1 lower bpp on MobileNetV2 ($94.8\%$ at 1.7 bpp, compared to $94.1\%$ at 2.8 bpp), and $0.2\%$ higher performance at 0.5 lower bpp on ShuffleNet ($92.0\%$ at 2.9 bpp, compared to $91.8\%$ at 3.4 bpp), while outperforming the full-precision model in both cases. Additional empirical studies are provided in Appendix D.

## 4.2 ImageNet

For the large-scale ImageNet classification task, we quantize the ResNet-18 and ResNet-50 models which allow for comparisons against LQ-Nets, DSQ, SLB, and BSQ. For both networks we train the weight parameters with SGD, a momentum of 0.9, a weight decay of $10^{-4}$ and a batch size of 256 which is distributed across 4 GPUs. We follow LQ-Nets in terms of data augmentation.

We train ResNet-18 for a total of 180 epochs: the first 120 are used for precision training and the last 60 for fine-tuning, and SGD has an initial learning rate of 0.1 which is decayed by 10 at epochs 45, 90, 150, and 165.

For ResNet-50, we start from a pre-trained, full-precision model and train for 100 epochs: allocating the first 60 for precision training and the remaining 40 for fine-tuning. An initial learning rate of 0.01 is decayed by 10 at epoch 30 for the precision training phase, while fine-tuning starts with the same learning rate of 0.01 which is decayed by 10 at epochs 15 and 30. Note that, as in the CIFAR-10 experiments, our training budget is considerably smaller than BSQ's, which also starts from a pre-trained model but has a budget of 180 additional epochs.

Results in Table 3 show that SMOL outperforms competing methods while achieving higher performance than the full-precision baselines. On ResNet-18, our method offers $0.6\%$ higher accuracy than LQ-Nets at 1.7 lower bpp ($69.9\%$ at 2.3 bpp compared to $69.3\%$ at 4.0 bpp), while also outperforming the full-precision model by $0.3\%$.

Table 3: Performance of different quantization methods on ResNet-18 and ResNet-50 models trained on ImageNet. * denotes results with Zero Precision Allocation.

| Method | ResNet-18 | | | ResNet-50 | | |
|---|---|---|---|---|---|---|
| | Average Bpp ↓ | Compression Ratio ↑ | Test Accuracy (%) | Average Bpp ↓ | Compression Ratio ↑ | Test Accuracy (%) |
| FP | 32.0 | 1.0 | 69.6 | 32.0 | 1.0 | 76.1 |
| SLB | 2.0/4 | 16.0 | 67.5 | | | |
| LQ-Nets | 3.0/3 | 10.7 | 68.2 | 3.0/3 | 10.7 | 74.2 |
| SMOL* | | | | 0.8/4 | 40.0 | 74.6 |
| SMOL* | 2.3/4 | 7.6 | 69.9 | 2.2/4 | 14.5 | 74.9 |
| LQ-Nets | 4.0/4 | 8.0 | 69.3 | 4.0/4 | 8.0 | 75.1 |
| DSQ | 4.0/4 | 8.0 | 69.6 | | | |
| BSQ | | | | 3.3/4 | 9.7 | 75.2 |
| BSQ | | | | 3.7/4 | 8.6 | 75.3 |
| SMOL* | | | | 4.1/4 | 7.0 | 77.1 |
| SMOL | 4.5/4 | 7.1 | 70.6 | | | |
| SMOL | 4.2/4 | 7.6 | 70.4 | 5.3/4 | 5.9 | 76.9 |

On ResNet-50, SMOL outperforms LQ-Nets at $2.2$ lower bpp, with an average number of bits of $0.8$ – lower than a binary network. With $4.1$ bpp, our method provides a $1.0\%$ improvement over the full-precision baseline, suggesting that the noise injection adopted by our method has additional regularizing effects that can further improve generalization. Performance by average precision plots for BSQ, SMOL, and LQ are given in Figure 4.

## 4.3 Image Generation with DCGAN

We train a DCGAN [39] model on the CIFAR-10 dataset to perform unconditional image generation at a resolution of $32 \times 32$. The generator is implemented as four transposed convolutional layers with batch norm and ReLU activations, while the discriminator consists of four strided convolutional layers with batch norm and LeakyReLU activations. Models are trained with the binary cross-entropy loss using Adam with a learning rate $= 2 \cdot 10^{-4}$ and $(\beta_1, \beta_2) = (0.5, 0.999)$. We only quantize the weights of the generator since it is the network used for deployment.

For BSQ, we first train a full-precision DCGAN for 30 epochs, quantize its weights to 8-bits, and conduct 100 epochs of precision learning with a pruning interval of 10 epochs and $\lambda = 2 \cdot 10^{-3}$. The models are then fine-tuned for 30 epochs, with a $10\times$ decayed learning rate. Similarly, for SMOL we train for a total of 160 epochs: 130 for precision training followed by 30 epochs for fine-tuning.

We randomly generate 10,000 samples for all methods, each assessed with Inception Score(IS) [42] and Fréchet Inception Distance (FID) [15]. As shown in Table 4, our method consistently outperforms BSQ, with higher generation quality at lower bpp, demonstrating generalization capability to the challenging task of quantizing GANs.

We drastically improve BSQ's FID from $37.1$ at $3.9$ bpp to $29.9$ at only $3.5$ bpp. For BSQ at $2.8$ bpp, SMOL* achieves $7.2$ better FID with only $1.0$ bpp. Even when evaluated at an extremely low bpp of $0.2$, our method still generates images with good quality.

## 4.4 Heterogeneous Precisions for Quantized Activations

In order to extend SMOL to train precisions for hidden activations, we introduce new trainable parameters to estimate the perturbation limit of activation outputs instead of weights. For an activation tensor $u$, we instantiate a new variable $s$ of the same shape which will be be trained jointly with the network's original parameters.

Table 4: Performance of BSQ and SMOL on image generation on CIFAR-10 with DCGANs.

| | DCGAN | | |
| Method | Avg. Bpp ↓ | Inception Score ↑ | FID ↓ |
|---|---|---|---|
| FP (30 epochs) | 32.0 | 4.70 | 38.6 |
| FP (160 epochs) | 32.0 | 5.67 | 25.8 |
| BSQ | 2.8 | 4.85 | 38.3 |
| SMOL* | 0.2 | 4.75 | 35.6 |
| SMOL* | 0.5 | 5.03 | 34.1 |
| SMOL* | 1.0 | 5.01 | 31.1 |
| SMOL | 1.7 | 5.02 | 34.1 |
| BSQ | 3.9 | 4.95 | 37.1 |
| SMOL* | 3.5 | 5.29 | 29.9 |

Table 5: Performance of PACT and SMOL when quantizing activations of a ResNet-20 trained on CIFAR-10.

| | ResNet-20 | |
| Method | Avg. Bpa ↓ | Test Acc. (%) |
|---|---|---|
| FP | 32.0 | 91.6 |
| PACT | 2.0 | 89.2 |
| SMOL-A | 1.8 | 90.3 |
| PACT | 3.0 | 91.4 |
| SMOL-A | 2.9 | 91.8 |
| SMOL-A | 2.5 | 91.4 |
| PACT | 5.0 | 91.6 |

Similarly to the weight quantization case described in Section 3, at each training iteration $t$ we sample a tensor $\epsilon^{(t)}$ with the same shape as $u$, where each component is drawn uniformly from $[-1, +1]$, and generate perturbed activations $u + \frac{M}{2} \cdot \sigma(s) \odot \epsilon^{(t)}$, which are used in place of $u$ throughout the next layers of the network. The scalar $M$ denotes the range of activation function used to compute $u$, *i.e.*, $M = 1$ for sigmoid and $M = 2$ for tanh activations, and is used to match the magnitudes of the activations and its perturbations. Since ReLU activations are unbounded, we use PACT [3] which clips values to $[0, \alpha]$ where $\alpha$ is a new trainable parameter – this yields $M = \alpha$. Once $s$ has been trained, we map its components to integer precisions which are used to quantize the activations $u$.

Table 5 shows the performance of a ResNet-20 trained on CIFAR-10 whose activations are quantized by PACT and SMOL (referred as SMOL-A): our method provides off-the-shelf improvements over PACT, offering higher accuracy at lower average bits per activation (bpa). All networks are trained for 200 epochs, following the training settings from Section 4.1.

## 4.5 Computational Efficiency

A key question is whether per-parameter precisions can result in inference energy cost reductions. The power required to multiply a 2-bit and 3-bit weight with a 4-bit activation is $2.41\times$ and $3.83\times$ higher than what is required for a 1-bit weight; and the latency is $1.91\times$ and $2.10\times$ higher, respectively. These numbers are estimated using ripple-carry adder based multiplier designs (which are suitable for low-precision operations) for the corresponding precision settings [2]. For the accumulation operations, we assume that the power and latency values are the same for different precision settings.

We take as an example the last convolutional layer of ResNet-20, for which BSQ assigns 2 bits for all parameters while our method assigns 1, 2, and 3 bits to 74%, 24.5%, and 1.5% of the parameters, respectively. In this case, the layer with precisions assigned by SMOL requires only 57.5% of computation power while improving latency by 36% compared to BSQ, which amounts to an energy cost reduction (power x latency) of 62.7%. Note that in real hardware designs, additional control overheads, *e.g.,* around 25% [34], are required to perform fine-grained mixed-precision operations. Although estimates, these suggest that a fine-grained precision allocation scheme can result in significant energy savings on hardware designed to support the corresponding arithmetic operations.

## 5 Discussion

With advances in hardware design and significant overparameterization of recent deep networks, learning compact model representations that can be leveraged by specialized hardware has become a key problem – one that is not restricted to the machine learning and systems communities.

Our work goes a step further in terms of network quantization by proposing SMOL, a modular and principled method that is able to learn heterogeneous precisions at the finest granularity, where different weights of the same model can be represented in distinct precisions regardless of how the weights are connected in the network's topology.

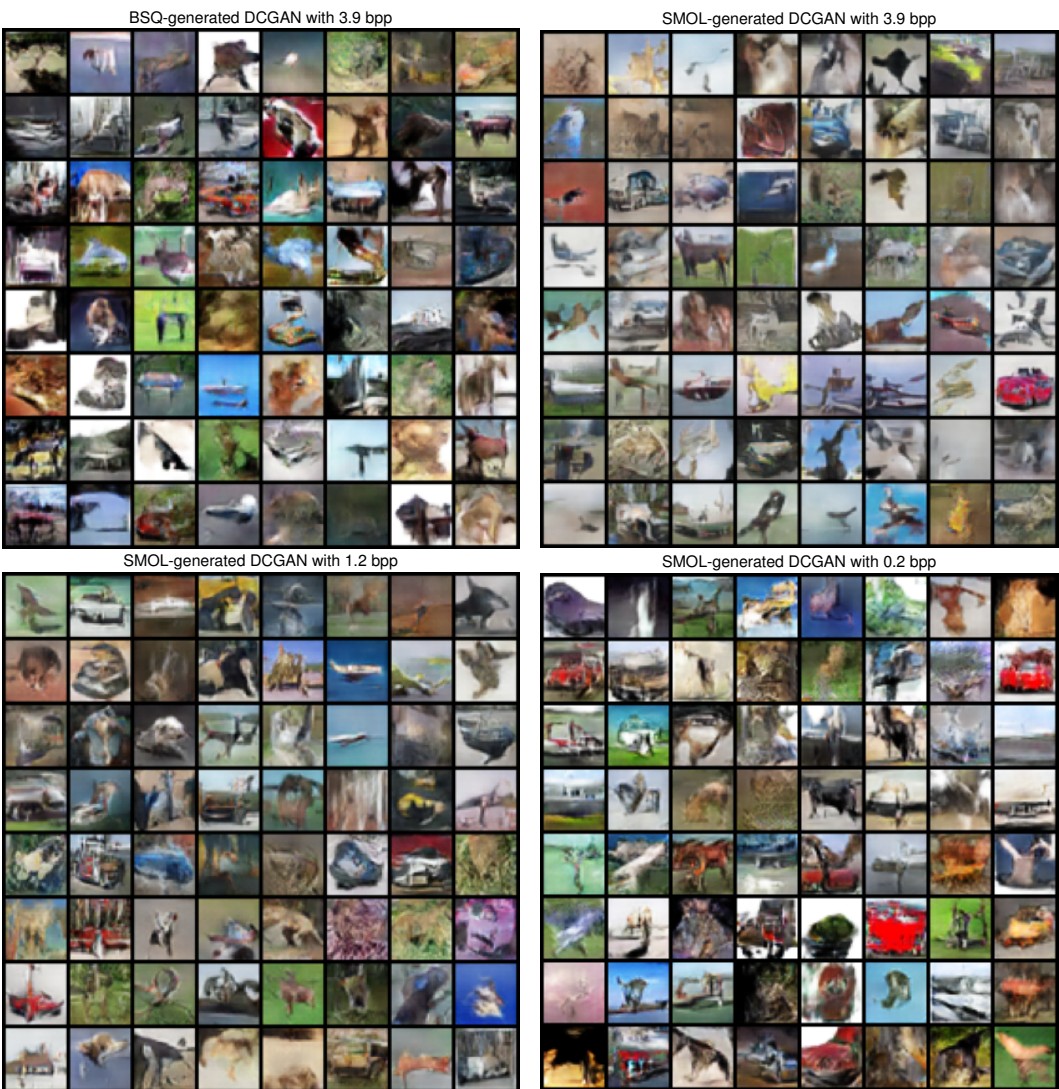

Figure 5: Image generations with a DCGAN trained on CIFAR-10, quantized with BSQ and SMOL.

Moreover, SMOL operates by introducing auxiliary variables to the target model, thus being agnostic to the network's topology, the choice of optimizer, or the underlying training setting, and hence it can be easily combined with other algorithms such as neural architecture search.

Our procedure to allocate precisions lies in the optimization of a trade-off between the model's performance and its cost – a cost function that can be chosen according to the underlying application or the user's concern, *e.g.,* computational or environmental cost.

Finally, our method offers an unification of pruning, binarization, and quantization, while at the same time achieving state-of-the-art results in image classification and image generation with quantized models. Our results show that adopting heterogeneous precisions yields highly compact models that not only perform well, but can also result in significant computational advantages when deployed on specialized hardware.

## Acknowledgments and Disclosure of Funding

This work was supported in part by the University of Chicago CERES Center. The authors have no competing interests.

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
