# Appendix

We present additional experiments and discussions in the following sections. Appendix A provides the derivation of our precision-training framework, including steps and approximations taken to arrive at our proposed regularized loss. Appendix B presents a discussion on how to accurately compare methods that use different bit values, which is crucial when comparing our method against BSQ [50] as it adopts unsigned bits and a mutual notion of precision must be established. Appendix C has additional details on our main experiments, including how $\lambda$ was chosen and how activations were quantized. Finally, Appendix D offers additional experimental results with the goal of studying key questions regarding our method, including whether it is able to assign precisions in a layer-wise fashion instead of per-parameter.

## A   Framework Derivation

### A.1   Preliminaries

We start by defining a function family $\mathcal{F}$ parameterized by $\Theta \subseteq \mathbb{R}^d$: for any $\theta \in \Theta$, we have a function $f_\theta \in \mathcal{F}$, $f_\theta : \mathbb{R}^{d_{in}} \to \mathbb{R}^{d_{out}}$ that is induced by $\theta$. In our practical setting, the variables $\theta$ denote a configuration for the parameters of neural network while $f_\theta$ represents the function that the network implements under such parameter values.

For the sake of convenience, we directly define a loss functional $L : \Theta \to \mathbb{R}$ that measures the (average) instant loss of a function $f_\theta \in \mathcal{F}$ induced by some $\theta \in \Theta$. For example, $L$ can be explicitly given by $L(\theta) = \frac{1}{N} \sum_{i=1}^{N} \ell(f_\theta(x_i), y_i)$, where $((x_i, y_i))_{i=1}^{N}$ is a fixed dataset and $\ell$ is an instant loss (*i.e.,* squared loss or cross-entropy).

Now, consider the problem of finding a parameter setting $\theta \in \Theta$ where each component $\theta_i$ is represented by exactly $p_i$ many (signed) bits given a bit-value map $(v_1, v_2, \dots)$ that assigns a scalar $v_j \in \mathbb{R}$ to each position $j \in \mathbb{N}$ in a bit string. In other words, we have

$$\theta_i = \sum_{j=1}^{p_i} b_{i,j} \cdot v_j \,,$$

where $b_{i,j} \in \{\pm 1\}$ is the $j$'th bit used to represent $\theta_i$ and $(v_j)_{j=1}^{P}$ is the bit-value map. W.l.o.g. we will assume that the bit-value map is (countably) infinite and that $\mathbb{V}^d \subseteq \Theta$, where $\mathbb{V} = \{\sum_{j=1}^{\infty} b_j \cdot v_j \mid b_j \in \{\pm 1\}\}$ is the set of all possible values that each parameter can assume.

We can then formalize the problem of finding a set of bit precisions $p \in \mathbb{N}^d$ and bit strings $b \in \{\pm 1\}^{d \times \infty}$ such that the total precision $\sum_{i=1}^{d} p_i$ is minimized and the parameter setting $\theta \in \Theta$ induced by $b$ achieves 'small loss':

**Definition A.1.** (Mixed Precision Problem)

$$\min_{p \in \mathbb{N}^d, \, b \in \{\pm 1\}^{d \times \infty}} \quad \sum_{i=1}^{d} p_i \qquad \text{s.t.} \quad L(V(b, p)) \le L^* + \delta \,, \; V(b_i, p_i) = \sum_{j=1}^{p_i} b_{i,j} \cdot v_j \qquad (3)$$

where $L^* = \min_{\theta \in \mathbb{V}^d} L(\theta)$ is the smallest achievable loss by $\mathcal{F}$ over $\mathbb{V}^d$ and $\delta \in \mathbb{R}^+$ is a scalar that captures tolerance in terms of suboptimality. Here, $V$ is the function that maps bit strings $b \in \{\pm 1\}^{d \times \infty}$ and precision values $p \in \mathbb{N}^d$ to parameter values $\theta \in \mathbb{V}^d$ element-wise:

$$V(b_i, p_i) = \sum_{j=1}^{p_i} b_{i,j} \cdot v_j \,.$$

The above problem poses a few obstacles: first, it involves a search over multiple binary values for each of the $d$ parameters of the network; second, the derivative w.r.t. each integer precision $p_i$ is undefined, making it ill-suited for gradient-based methods.

Alternatively, we can directly optimize $\theta \in \mathbb{V}^d$ under the constraint that its finite-precision representation is used to evaluate the loss $L$:

**Definition A.2.** (Mixed Quantization Problem)

$$\min_{p \in \mathbb{N}^d, \, \theta \in \mathbb{V}^d} \quad \sum_{i=1}^{d} p_i \qquad \text{s.t.} \quad L(Q(\theta, p)) \leq L^* + \delta \tag{4}$$

where $Q$ maps $\theta \in \mathbb{V}$ to a finite-precision representation of itself with exactly $p$ many bits. To define $Q$ precisely we will first introduce a notion of inverse for $V$ (a 'notion' since $V$ is not generally bijective).

For any $\theta_i \in \mathbb{V}$, we let $V^{-1}(\theta_i)$ denote a setting $(b_i, p_i)$, with $p_i \in \mathbb{N}$ and $b_i \in \{\pm 1\}^\infty$ such that $V(b_i, p_i) = \theta_i$ and, moreover, for any $(b_i', p_i')$ such that $V(b_i', p_i') = \theta_i$, it follows that $p_i' \geq p_i$. That is, $V^{-1}$ maps each element $\theta_i \in \mathbb{V}$ to a precision-bitstring tuple that perfectly represents $\theta_i$ using the least number of bits. We also define $V_p^{-1}(\theta_i) = p_i$ and $V_b^{-1}(\theta_i) = b_i$ where, as before, $V^{-1}(\theta_i) = (b_i, p_i)$.

Then, we can define $Q$ by $Q(\theta_i, p_i) = \sum_{j=1}^{p_i} V_b^{-1}(\theta_i)_j \cdot v_j$, *i.e.*, it is the value induced by the *shortest* bit string that represents $\theta$, when considering only its first $p_i$ bits. Note that this yields $Q(\theta_i, p_i) = V(V_b^{-1}(\theta_i), p_i)$, which will be a key property in the proofs below.

While it might be tempting to see $Q$ as a quantization mapping, it not always acts as such. For example, consider the value map given by $v_j = 2^{1-j}$, where $\theta_i = 1$ can be represented by $p^{(1)} = 1$, $b_i^{(1)} = (1, 1, 1, 1, \dots)$, by $p^{(2)} = \infty$, $b_i^{(2)} = (1, -1, 1, 1, \dots)$, and so on. In this case, we could let $V^{-1}(1) = (b^{(1)}, p^{(1)})$, but not $V^{-1}(1) = (b^{(2)}, p^{(2)})$. For the former, we would have $Q(1, 2) = \frac{3}{2}$, which is *not* a quantization – in the common sense – of $\theta_i = 1$ under the considered value map.

With this definition of $Q$ we can show that the optimization problems in Equations (3) and (4) are equivalent (under $V$ and $V^{-1}$).

**Lemma A.3.** *For any setting* $(b, p)$ *that satisfies* $L(V(b, p)) \leq L^* + \delta$ *in Equation* (3)*, we have that* $(\theta = V(b, p), p)$ *satisfies* $L(Q(\theta, p)) \leq L^* + \delta$ *in Equation* (4)*.*

*Proof.* By definition, we have:

$$\begin{aligned}
Q(\theta, p) &= Q(V(b, p), p) \\
&= V\left(V_b^{-1}(V(b, p)), p\right) \\
&= V(b, p) \\
&= \theta
\end{aligned} \tag{5}$$

Therefore, the parameter setting induced by $V(p, b)$ in Equation (3) will be the same as the one induced by $Q(\theta, p)$ in Equation (4), and hence the losses will be the same and satisfiability follows. $\square$

**Lemma A.4.** *For any setting* $(\theta, p)$ *that is a minimizer of the problem in Equation* (4)*, we have that* $(b = V_b^{-1}(\theta), p)$ *satisfies* $L(V(b, p)) \leq L^* + \delta$ *in Equation* (3)*.*

*Proof.* Since $(\theta, p)$ minimizes $\sum_{i=1}^{d} p_i$ by assumption, we have that no $\theta_i$ can be represented with less than $p_i$ many bits and hence $p = V_b^{-1}(\theta)$. Therefore

$$\begin{aligned}
V(b, p) &= V(V_b^{-1}(\theta), p) \\
&= V(V_b^{-1}(\theta), V_p^{-1}(\theta)) \\
&= V(V^{-1}(\theta)) \\
&= \theta
\end{aligned} \tag{6}$$

Therefore, the parameter setting induced by $Q(\theta, p) = \theta$ in Equation (4) will be the same as the one induced by $V(b, p)$ in Equation (3), and hence the losses will be the same and satisfiability follows. $\square$

With the two Lemmas we can prove that the two problems are indeed equivalent:

**Corollary A.5.** *The optimization problems in Equations* (3) *and* (4) *are equivalent under* $V$ *and* $V^{-1}$. *That is, for any* $(b, p)$ *that minimizes* (3), *we have that* $(\theta = V(b, p), p)$ *minimizes* (4), *and for every* $(\theta, p)$ *that minimizes* (4) *we have that* $(b = V_b^{-1}(\theta), p)$ *minimizes* (3). *Moreover, the induced function* $f_\theta$ *will be the same in all cases.*

*Proof.*

- (4) $\rightarrow$ (3): assume for the sake of contradiction that $(b = V_b^{-1}(\theta), p)$ is suboptimal in (3), then there exists a satisfiable $(b', p')$ with $\|p'\| < \|p\|$, but from Lemma A.3 it follows that $(\theta = V(b', p'), p')$ is satisfiable in (4), which contradicts the optimality of $(\theta, p)$.

- (3) $\rightarrow$ (4): assume for the sake of contradiction that $(\theta = V(b, p), p)$ is suboptimal in (4), then there exists a satisfiable $(\theta', p')$ with $\|p'\| < \|p\|$, but from Lemma A.4 it follows that $(b = V_b^{-1}(\theta', p'), p')$ is satisfiable in (3), which contradicts the optimality of $(b, p)$.

$\square$

## A.2 Stochastic Approximation

We now present a sequence of steps, involving approximations, that yield our final method.

1. We can first write
$$Q(\theta, p) = \theta - (\theta - Q(\theta, p)) = \theta + R(\theta, p)$$
which allows us to show that
$$R(\theta_i, p_i) = \left( \sum_{j=p+1}^{V_p^{-1}(\theta_i)} V_b^{-1}(\theta_i)_j \cdot v_j \right) + \left( \sum_{j=V_p^{-1}(\theta_i)}^{p+1} V_b^{-1}(\theta_i)_j \cdot v_j \right)$$
by relying on the definition of $Q$ and $V^{-1}$ alone.

2. Next, we consider the following assumption: if $(\theta_i, p_i)$ is satisfiable, then $(\theta_i, p_i')$ is satisfiable for any $p_i' > p_i$ (i.e. assigning more bits to a parameter preserves its satisfiability). This assumption can be more simply framed as
$$L(Q(\theta, p)) < L^* + \delta \implies L(Q(\theta, p')) < L^* + \delta, \quad \forall i\ p_i' > p_i \,,$$
and is the core assumption for our method.

3. Using the definition of $R$ and w.l.o.g. denoting $Q(\theta, p)$ simply by $\theta'$, we get from the above equation that
$$L(\theta') < L^* + \delta \implies L(\theta' + R(\theta', p')) < L^* + \delta, \quad \forall i\ p_i' > V_p^{-1}(\theta_i')$$
*i.e.,* using the trinary system, this means that if a bitstring $(1, 1, 0, 0, 0, \dots)$ is satisfiable, then any bitstring $(1, 1, \pm 1, \pm 1, \pm 1, \dots)$ will also be satisfiable. Now, we can frame the following optimization problem which has infinitely many constraints:
$$\min_{p \in \mathbb{N}^d,\ \theta \in \mathbb{V}^d} \quad \sum_{i=1}^d p_i \qquad \text{s.t.} \quad L(\theta + R(\theta, p')) \leq L^* + \delta, \quad \forall i\ p_i' > p_i = V_p^{-1}(\theta_i) \quad (7)$$

4. Note that $R(\theta_i, p_i')$ includes $V_b^{-1}(\theta_i)$ which admits any configuration for all bits after $V_b^{-1}(\theta_i)$-pth one due to the fact that $Q$ operates via performs truncation. We further assume that $\{R(\theta_i, p_i') \mid p_i' > V_p^{-1}(\theta_i)\}$ is dense in some interval $I(\theta_i, p_i) = [l, u] \subset \mathbb{V}$.
   We then frame a stochastic approximation for the problem above, which is designed by sampling from the infinitely many constraints that were introduced:
$$\min_{p \in \mathbb{N}^d,\ \theta \in \mathbb{V}^d} \quad \sum_{i=1}^d p_i \qquad \text{s.t.} \quad \frac{1}{K} \sum_{k=1}^K L(\theta + \epsilon^{(k)}) \leq L^* + \delta, \quad \forall (i, k),\ \epsilon_i^{(k)} \sim \mathbb{P}(I(\theta_i, p_i))$$
$$(8)$$

5. Next, we pose the constraints using Lagrange multipliers to yield a regularized objective:

$$\min_{p\in\mathbb{N}^d,\,\theta\in\mathbb{V}^d} \quad \frac{1}{K}\sum_{k=1}^{K} L(\theta+\epsilon^{(k)})+\lambda\|p\|, \quad \forall(i,k),\; \epsilon_i^{(k)}\sim\mathbb{P}(I(\theta_i,p_i)) \tag{9}$$

6. Note that the derivation so far is agnostic to the underlying value map. We now assume the value map is given by $v_j = 2^{1-j}$, where we get $\mathbb{P}(I(\theta_i,p_i))=\mathcal{U}(\pm 2^{1-p_i})$ under a bit distribution of $\mathcal{U}(\pm 1)$ for all bits after the $V_p^{-1}(\theta_i)$-th one:

$$\min_{p\in\mathbb{N}^d,\,\theta\in\mathbb{V}^d} \quad \frac{1}{K}\sum_{k=1}^{K} L(\theta+\epsilon^{(k)})+\lambda\|p\|, \quad \forall k,\; \epsilon_i^{(k)}\sim\mathcal{U}^d(\pm 2^{1-p_i}) \tag{10}$$

7. To allow for gradients to flow through the sampling procedure, we apply the reparameterization trick and a continuous relaxation on $p$:

$$\min_{p\in\mathbb{R}_+^d,\,\theta\in\mathbb{V}^d} \quad \frac{1}{K}\sum_{k=1}^{K} L(\theta+2^{1-p}\odot\epsilon^{(k)})+\lambda\|p\|, \quad \forall k,\; \epsilon^{(k)}\sim\mathcal{U}^d(\pm 1) \tag{11}$$

8. Finally, we reparameterize both $p$ and $\theta$ to yield variables that can be optimized over the reals – although not necessary, this results in a procedure that does not require variables to be projected to a domain after each update.
We define $s = \sigma^{-1}(2^{1-p})$, $\theta = \tau(w)$, and frame the problem as optimization over $(s,w)$ instead, with the advantage that, unlike $(p,\theta)$, they both belong to $\mathbb{R}^d$:

$$\min_{s,w\in\mathbb{R}^d} \quad \frac{1}{K}\sum_{k=1}^{K} L(\tau(w)+\sigma(s)\odot\epsilon^{(k)})+\lambda\|\log_2(1+e^{-s})\|, \quad \forall k,\; \epsilon^{(k)}\sim\mathcal{U}^d(\pm 1) \tag{12}$$

where ideally we have $\tau:\mathbb{R}\to\mathbb{V}$ be such that $\tau(\mathbb{R})=\mathbb{V}$.

This concludes the derivation of our regularized loss. Note that instead of adopting the reparameterization $\theta=\tau(w)$, we can directly optimize $\theta$ while guaranteeing that $\theta\in\mathbb{V}$: this can be easily achieved by projecting the parameters $\theta$ to an interval, for example to $[-1,+1]$ when using signed bit representations.

## B  Bit Counting: Implicit vs Explicit Sign Bits

The process of quantizing data is heavily dictated by whether bits are chosen to be signed (taking values $\{+1,-1\}$) or unsigned (values $\{0,1\}$), and comparing methods that use different settings might require adopting notions of 'precision' that diverge from the ones used in their original papers.

Most methods on quantizing neural networks [53, 51, 10] – ours included – adopt signed bits, which is often done implicitly by splitting the $[-1,+1]$ interval uniformly in $2^p$ bins to extract the values that quantized data can assume. In this case, assigning a value of $2^{1-i}$ to each $i$-th bit results in a bitstring $(b_1,b_2,\ldots,b_p)$ being mapped to $\sum_{i=1}^{p} b_i\cdot 2^{1-i}$, a variable that can assume $2^p$ different values uniformly distributed over an interval centered at the origin *i.e.,* $[-M,+M]$ for positive $M$.

On the other hand, adopting unsigned bits results in the possible values for $\sum_{i=1}^{p} b_i\cdot 2^{1-i}$ to be uniformly distributed over a $[0,M']$ interval instead (for positive $M'$), hence bitstrings can only be mapped to non-negative values. An additional 'sign bit' can be adopted to allow the representation of negative numbers: in particular, points are encoded to unsigned bitstrings of the form $(b_s,b_1,\ldots,b_{p'})$, which are then mapped to $s(b_s)\cdot\sum_{i=1}^{p'} b_i\cdot 2^{1-i}$ where $s(b_s)=+1$ if $b_s=1$ and $s(b_s)=-1$ if $b_s=0$: note that such mapping allows for a total of $2^{p'+1}$ different outputs. In this case the term 'precision' $p$ can be used to either denote $p'+1$ (the total number of bits used to encode each point) or $p'$ (number of bits except for the 'sign bit' $b_s$). However, when comparing a method that relies on such representation to one that adopts signed bits instead, we must count the additional 'sign bit' *i.e.,* $p=p'+1$, as in this case both methods can represent exactly $2^p$ values given a precision of $p$ bits.

When considering results reported by BSQ [50], we account for an additional 'sign bit' that is implicitly used when quantizing full-precision weights with bitstrings. More specifically, once a

model has been fine-tuned, BSQ converts its full-precision weights $W_s \in \mathbb{R}^d$ to binary tensors $W_p^{(0)}, \ldots, W_p^{(n-1)}, W_n^{(0)}, \ldots, W_n^{(n-1)}$, each $\in \{0, 1\}^d$, where $(W_p^{(b)})_{b=0}^{n-1}$ encodes the positive components of $W_s$ while $(W_n^{(b)})_{b=0}^{n-1}$ encodes the absolute value of its negative components.

These binary tensors are then mapped to quantized weights $W_{final} = \frac{1}{2^n-1} \sum_{b=0}^{n-1} (W_p^{(b)} - W_n^{(b)}) 2^{n-b-1}$, whose components can take values $\frac{1}{2^n-1}\{0, 1, \ldots, 2^n - 1\}$ by setting $W_n^{(b)} = 0$ and enumerating all configurations for $(W_p^{(b)})_{b=0}^{n-1}$, while setting $W_p^{(b)} = 0$ and considering all configurations for $(W_n^{(b)})_{b=0}^{n-1}$ yields the values $\frac{-1}{2^n-1}\{0, 1, \ldots, 2^n - 1\}$: in total, we have that elements of $W_{final}$ can assume $2^{n+1} - 1$ distinct values ($2^n - 1$ positive, $2^n - 1$ negative, and 0). An implicit 'sign bit' is also used in the forward computation of quantized models, which maps weights $W$ to $\frac{1}{2^n-1} \cdot \text{sign}(W) \odot \lceil |W| \cdot (2^n - 1) \rceil$: the implicit representation of $n$ bits does not encode the sign of the weights, which is recovered after quantizing its absolute value over $2^n$ bins.

To summarize, BSQ uses unsigned bit representations to encode weights but without an explicit 'sign bit': the weight signs are either bypassed and hence not encoded (as in the forward pass), or are implicitly encoded by allocating two bitstrings for each weight (as when converting full-precision weights to bitstrings)[1] – where which one is used is dictated the the weight's sign.

To account for the mismatch in how many values can be represented with $p$ many bits by BSQ compared to our method, we account for one more bit when reporting BSQ results, which designates the implicit 'sign bit' in addition to the unsigned $n$-bit representations.

## C   Experimental Details

**Values for $\lambda$:**   When training ResNet-20 networks on CIFAR-10, we used $\lambda \in \{10^{-6}, 7 \cdot 10^{-7}, 5 \cdot 10^{-7}\}$ when running SMOL, where larger values for $\lambda$ result in less bits per parameter: $\lambda = 10^{-6}$ resulted in a network with 2.1 bpp while $\lambda = 5 \cdot 10^{-7}$ yielded a model with 2.8 bpp. For MobileNetV2 we used $\lambda \in \{10^{-7}, 2 \cdot 10^{-7}\}$, while for ShuffleNet we adopted $\lambda \in \{5 \cdot 10^{-8}, 7 \cdot 10^{-10}\}$

For our ImageNet experiments, we adopted $\{10^{-6}, 10^{-8}\}$ and $\{10^{-7}, 10^{-8}\}$ as values for $\lambda$ when training ResNet-18 and ResNet-50 networks, respectively. Different values for $\lambda$ were required (compared to the ones we used for CIFAR-10) due to the ImageNet ResNets having considerably more parameters than the ResNet-20 model we adopted for CIFAR-10: note that our regularizer is a sum over all precisions instead of, for example, an average.

Finally, for image generation we used $\lambda \in \{2 \cdot 10^{-5}, 10^{-5}, 5 \cdot 10^{-6}, 2 \cdot 10^{-6}, 10^{-6}\}$, where the most compact model (0.2 bpp) was achieved with $\lambda = 2 \cdot 10^{-5}$ along with zero precision allocation. In most cases we saw a decrease of between 1 and 2 bpp when enabling zero precision allocation: training a DCGAN with $\lambda = 10^{-5}$ results in 0.5 bpp with zero precision allocation and 1.7 without it.

**Quantized activations for ImageNet Experiments:**   We follow BSQ [50] and replace all ReLU modules by ReLU6 throughout the network, resulting in outputs constrained to the $[0, 6]$ interval. Activations are not quantized prior to fine-tuning *i.e.,* precision training with SMOL uses full-precision activations, which are only quantized once precisions have been allocated and fine-tuning has started. All our experiments that quantize activations consider 4-bit precisions *i.e.,* the activation tensors are quantized using $2^4 = 16$ different values.

Also following BSQ, we use straight-through estimators to allow for gradient flow through the activation quantization procedure. The quantization values are uniform over half of the activation range – typically $[0, 3]$, which results in the full-precision activations being quantized to one of the 16 values in the set $\{0, \frac{1}{5}, \frac{2}{5}, \ldots, \frac{14}{5}, \frac{15}{5}\}$.

---

[1]These refer to *bit_STE* and *finetune_to_bit* in the official github repository: https://github.com/yanghr/BSQ/tree/9f09b6284d264ce07445e42ddd18764c7b3d0405

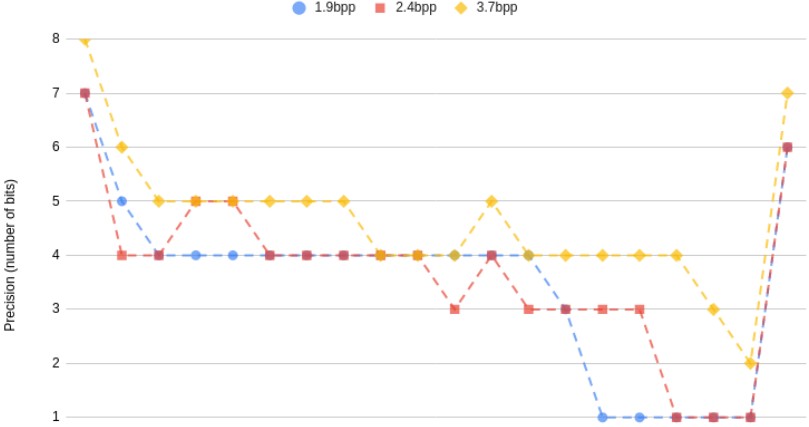

Figure 6: Layer-wise precisions allocated by SMOL-L on ResNet-20 models trained on CIFAR-10. The x-axis denotes the layer indices: leftmost points denote earlier layers while rightmost denote layers in the later stages of ResNet, which are closer to the final fully-connected layer.

# D   Additional Experimental Questions

## D.1   What Precision does SMOL Allocate to Each Layer?

In Section 4.1 we present results for SMOL when training layer-wise precisions on a ResNet-20 trained on CIFAR-10 (SMOL-L in Table 1), where higher performance under lower bpp is achieved compared to BSQ.

Figure 6 shows the precisions allocated to each layer of the ResNet-20 under three different values for $\lambda$: in all cases, later convolutional layers are assigned low precisions compared to earlier stages of the network. The first convolution, along with the fully-connected layer at the end of the network, are assigned significantly higher precision than other layers.

## D.2   More Precision Training or More Fine-tuning?

In our main experiments we allocate roughly half of the training budget to precision training and the other half to fine-tuning *i.e.,* further optimization of the weight parameters while applying quantization and adopting straight-through estimators. Here we show how the performance and size (measured in average number of bits per parameter) of the final model behaves under different ratios for the number of epochs allocated to precision training over the total training budget.

We use the settings described in Section 4.1, training models with $\lambda = 7 \cdot 10^{-7}$ but changing what how many of the total 650 epochs are used for precision training – the remaining ones are always allocated to fine-tuning the model.

Results are shown in Figure 7: there is significant change in the model's bpp when the fraction of epochs allocated to precision training is lower than $\frac{250}{650}$, indicating that SMOL requires between 200 and 250 epochs to fully optimize the auxiliary variables $s$. For ratios larger than $\frac{250}{650}$ our method yields networks with similar bits-per-parameter (around 2.5 bpp), but the test accuracy starts to decrease as the ratio increases due less epochs being used to fine-tune the model. The best performance is achieved by allocating 350 epochs for precision training, which results in the remaining 300 epochs being assigned for fine-tuning.

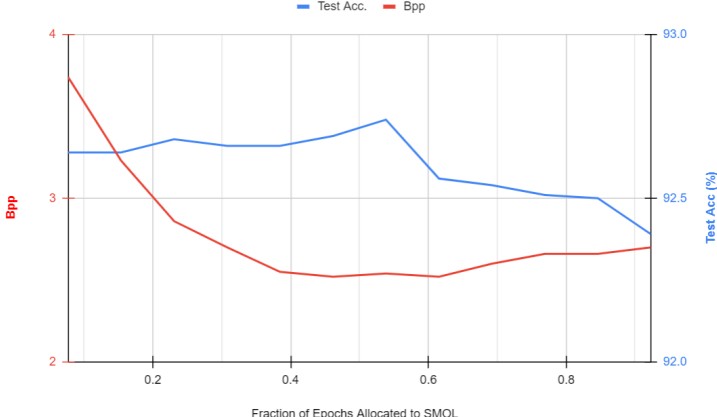

Figure 7: Performance and average precision of ResNet-20 trained on CIFAR-10 with SMOL when allocating different number of epochs to precision training versus fine-tuning. Curve shows increases of 50 epochs over a total of 650 epochs.

Table 6: Comparison between a ResNet-20 trained on CIFAR-10 with SMOL and its performance when re-trained, either when the per-weight precision assignments are maintained or randomly permuted (shuffled).

| Method | ResNet-20 | | |
| --- | --- | --- | --- |
| | Average Bpp $\downarrow$ | Compression Ratio ($\times$) $\uparrow$ | Test Accuracy (%) |
| Original (SMOL) | 2.5 | 12.8 | 92.8 |
| Re-trained with same precisions | 2.5 | 12.8 | 91.5 |
| Re-trained with shuffled precisions | 2.5 | 12.8 | 89.3 |

## D.3 Does SMOL Learn Structured Precision Maps?

To evaluate whether the per-parameter precision assignments learned by SMOL have an underlying structure or not, we set up an additional experiment which aims to re-train and evaluate a network after destroying any possible structure in its precision map. In particular, we start from a ResNet-20 model that has been fully trained and fine-tuned by SMOL, and proceed to re-train it under two different settings.

For the first, we re-initialize the weight parameters (using a different random seed) but fully preserve the precision assignments generated by SMOL *i.e.,* each $i$-th weight $w_i$ with precision $p_i$ is randomly re-initialized ($w_i \sim \mathcal{D}_{init}$), but $p_i$ is kept the same. As for the second setting, we also apply a random permutation on the precision tensor of each layer *i.e.,* weights $w_i$ and $w_j$ of the same layer have their precisions $p_i$ and $p_j$ swapped – with the goal of destroying any possible structure that the tensor $p$ might have.

We then re-train the two networks using the same training settings adopted in Section 4.1, but skipping precision training and hence allocating a budget of 300 epochs to weight training under quantization and straight-through estimators, akin to fine-tuning.

Table 6 and Figure 8 present results: randomly permuting (shuffling) the precisions yields in significantly lower performance (89.3% compared to 91.5%), suggesting that the precision assignments generated by SMOL do have an underlying structure, which can be some form of per-layer organization *i.e.,* allocating high or low precision to all elements in the same convolutional filter, or cross-layer structure *i.e.,* feature maps (layer outputs) have low or high precision connections to both the previous and the next layer. We leave further the characterization of such structure to future work.

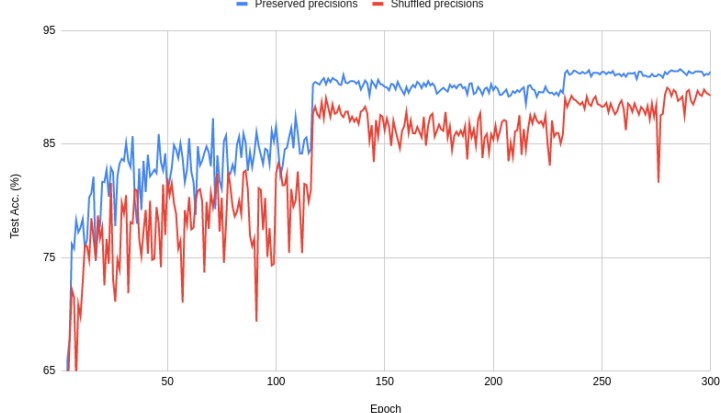

Figure 8: Generalization performance when re-training networks, with and without randomly permuted (shuffled) precisions. The model whose precisions have been shuffled has visibly lower performance across all stages of training and achieves significantly lower final performance.

### D.4  Can SMOL Quantize Transformer Models?

Since our results in Section 4 focus on CNN-like networks like ResNets and MobileNetV2, an immediate question is whether SMOL is able to efficiently quantize architectures of different families, for example Transformers which do not use convolutions and whose main component is the attention mechanism.

To answer this question, we run preliminary experiments where we train a standard encoder-decoder Transformer with 6 attention blocks on the neural machine translation IWSLT'14 German to English task, where we quantize all weights except for the layer normalization parameters. A total of 50 epochs were used for training: for SMOL, the first 30 were used to train the precisions and the remaining 20 for fine-tuning.

Table 7: Performance of SMOL when quantizing a Transformer on IWSLT'14.

| Method | Transformer | |
| | Avg. Bpp ↓ | BLEU Score ↑ |
| --- | --- | --- |
| FP | 32.0 | 34.9 |
| SMOL | 3.9 | 34.7 |
| SMOL | 5.8 | 34.9 |

Results in Table 7 show that SMOL can match the performance of the full-precision model at $5.8$ bpp, which amounts to a $5.5\times$ compression ratio. At a lower bpp of $3.9$ ($8.2\times$ ratio), the BLEU score is lower by only $0.2$. Note that we use the same training budget that is commonly employed when training a full-precision model for this task, which is likely suboptimal for quantization. Therefore, although promising, these results are preliminary and can likely be improved by simply training for more epochs.