# OpenReview forum: "Not All Bits have Equal Value: Heterogeneous Precisions via Trainable Noise"
_NeurIPS.cc/2022/Conference — NeurIPS 2022 Accept_

### Official Review · Reviewer_JFRS · 2022-07-07

**Rating:** 8
**Confidence:** 4
**Soundness:** 3 good
**Presentation:** 4 excellent
**Contribution:** 3 good

**Summary:**

The authors propose a technique for dynamically allocating quantization precision for neural networks. Their approach works by randomly generating a uniform noise tensor of the same shape as the parameters, and rescaling the noise by a per-parameter scalar. Based on the tolerable perturbation, it is possible to deduce what the appropriate precision should be. They provide results on CIFAR and ImageNet with standard models.


**Questions:**

See previous section.

I'm happy to increase score if you make some progress on them.

=== Post rebuttal:

these have mostly been answered and I'm happy. Score will be increased.

**Limitations:**

The authors check this box on the checklist but I am not convinced there is a clear description of the methods disadvantages. It would be nice to add a summary (I certainly won't penalise you).

==== Post rebuttal

discussed below

**Strengths And Weaknesses:**

I like this work, although I have a few concerns. It's well written and I like the approach overall.

Strengths

I do find the approach intuitive and satisfying. Unifying quantization with pruning is interesting from a theoretical perspective. I do think this work could inspire many new ideas in this community.

The results are good too, although it would be good if the main paper had more. Too many interesting results are shoved in the appendix (more on this below)

The approach is simple enough to be relatively easy to implement

The paper is well written. Figure 1 is very good :-)

Weaknesses

The most obvious issue with this work is that main setup presented is per-parameter precisions. This is not a realistic setup in the real world, and is not hardware efficient. The authors do consider layerwise precisions in the appendix (although more experiments would be desirable). Why are these not in the main paper? You have half a page left!

The evaluation only considers ResNets. Some other standard models would be useful. For example MobileNet/EfficientNet family are a good one to check, since they are difficult to quantize. Following this line of thought, what about other models (Transformers?)

Not enough discussion on activation quantization is given in the main text. Some is provided in the appendix. Quantizing to 4 bits is fine, although it might be interesting to consider other values. Some discussion of how to extend the proposed approach to activations would be nice to read (note: I am not actually asking you to do it -- just some comments).

---

> ### Author Response · Authors · 2022-08-02
> **Response to Reviewer JFRS (1/2)**
>
> **Per-parameter precision setup**: We have added a discussion in the above response to all reviewers on the applicability of per-parameter precisions. Specialized hardware designs (recent research results) point to the potential for future hardware to support fine-grained heterogeneous precision arithmetic operations, thereby achieving real world speedups on models produced by SMOL. We also have expanded the per-layer precision results (details in the response to all reviewers) and will add them to the main paper as suggested.
>
> **MobileNet/EfficientNet**: We have run additional experiments on MobileNetV2, EfficientNet, and ShuffleNet on CIFAR to evaluate how SMOL and BSQ compare on architectures that should be more difficult to quantize. We follow the same experimental setup as described in Section 4.1, training SMOL for 650 epochs and BSQ for 1000 epochs (since it requires pre-training and consists of 3 phases). The table below presents the test accuracies with the corresponding average bits per parameter (bpp) in parentheses.
>
> \\begin{array} 	{|r|r|} 	\\hline
> \\text{Method} 	&	\\text{MobileNetV2} 	& 	\\text{EfficientNet} 	& 	\\text{ShuffleNet}	\\\\ \\hline
> \\text{Full Precision}	& 	94.4 (32.0)	& 	92.7 (32.0)	& 	90.7 (32.0)	\\\\ \\hline
> \\text{BSQ} 		& 	94.1 (2.8)	& 	93.3 (2.6)	& 	91.8 (3.4)	\\\\ \\hline
> \\text{SMOL} 		& 	94.5 (1.5)	& 	93.1 (1.7)	& 	91.6 (1.8)	\\\\ \\hline
> \\text{SMOL} 		& 	94.7 (1.6)	& 	93.4 (2.6)	& 	92.0 (2.9)	\\\\ \\hline
> \\end{array}
>
> SMOL is able to match or outperform the full precision baseline with less than 1.8 bpp for all networks, while also outperforming BSQ. The improvement over BSQ is more significant on MobileNetV2, where SMOL offers 0.6% higher accuracy (94.7 compared to 94.1) at a 1.2 lower average bit per parameter (1.6 compared to 2.8).
>
> **Transformers**: We have also run preliminary experiments to evaluate whether SMOL is capable of quantizing Transformer models while preserving the original performance. We have trained a standard encoder-decoder Transformer (6 attention blocks each) on the neural machine translation IWSLT'14 German to English task, where we quantize all parameters except for the affine transformation of LayerNorm layers. A total of 50 epochs were used for training: for SMOL, the first 30 were used to train the precisions and the remaining 20 for fine-tuning.
>
> \\begin{array} 	{|r|r|} 	\\hline
> \\text{Method} 	&	\\text{BLEU} 		& 	\\text{Avg. Bits per Parameter}	& \\text{Compression Ratio}	\\\\ \\hline
> \\text{Full Precision}	& 	34.9 		& 	32.0		& 1 \\times			\\\\ \\hline
> \\text{SMOL} 		& 	34.7 		& 	3.9		& 8.2 \\times			\\\\ \\hline
> \\text{SMOL} 		& 	34.9 		& 	5.8		& 5.5 \\times			\\\\ \\hline
> \\end{array}
>
> SMOL is able to quantize the Transformer down to 5.8bpp while preserving its original performance of 34.9 BLEU. At the cost of 0.2 BLEU score, SMOL is capable of increasing the compression ratio from 5.5x to 8.2x. We believe that further compression can be achieved by training for longer and note that while promising, these results are preliminary and can likely be improved.

---

> > ### Author Response · Authors · 2022-08-02
> > **Response to Reviewer JFRS (2/2)**
> >
> > **Activation Quantization**: To assess whether SMOL can be generalized to quantize activations instead of weights, we introduce the new variables $s$ to each activation instead of each parameter and train them following SMOL. More specifically, instead of using perturbed weights $w + \sigma(s) \odot \epsilon$ (as in the parameter quantization setting), we use perturbed activations $u +\sigma(s) \odot \epsilon$, where $u$ is an activation (e.g. the output of a ReLU). Following our original method, this allows us to train $s$ and then map them to per-activation precisions. Afterwards, the network is fine-tuned while the activations are quantized instead of perturbed. We train a ResNet-20 on CIFAR for 200 epochs, allocating 100 epochs for precision training and the last 100 for fine-tuning, where only the activations (and not the weights) are quantized in order to isolate the effects of activation quantization.
> >
> > \\begin{array} 	{|r|r|} 	\\hline
> > \\text{Method} 	& \\text{Test Acc.}	& \\text{Avg. Bits per Activation} 	& \\text{Compression Ratio} \\\\ \\hline
> > \\text{Full Precision}	& 91.6		& 32.0				& 1.0 \\times			\\\\ \\hline
> > \\text{PACT} 		& 89.2		& 2.0				& 16.0 \\times		\\\\ \\hline
> > \\text{PACT} 		& 91.4		& 3.0				& 10.7 \\times		\\\\ \\hline
> > \\text{PACT} 		& 91.6		& 5.0				& 6.4 \\times			\\\\ \\hline
> > \\text{Ours} 		& 90.3		& 1.8				& 17.8 \\times		\\\\ \\hline
> > \\text{Ours} 		& 91.4		& 2.5				& 12.8 \\times		\\\\ \\hline
> > \\text{Ours} 		& 91.8		& 2.9				& 11.0 \\times		\\\\ \\hline
> > \\end{array}
> >
> > Our activation quantization approach significantly outperforms PACT, yielding 90.3% acc at 17.8x compression while PACT offers 89.2% at a worse compression rate of 16x. At a compression of roughly 11x, our method achieves 0.4% higher accuracy than PACT (91.8 compared to 91.4) and outperforms the full precision baseline (91.6%). These results show that directly applying our proposed framework (Appendix A) to activations yields a competitive method that offers immediate improvements over previous approaches.
> >
> > **Limitations / Disadvantages**: Our method’s limitations include 1) specialized hardware is needed to fully leverage per-parameter precisions, 2) the final precisions depend on the lambda hyperparameter, hence training a model with a specific final bpp as goal might require exploring multiple values for lambda, 3) although SMOL requires less training phases than BSQ (2 instead of 3, since it does not require pre-training), it still requires fine-tuning once the final precisions are assigned. We will expand the discussion on limitations in the paper.

---

> > > ### Comment · Reviewer_JFRS · 2022-08-07
> > > **Response to authors**
> > >
> > > Thanks for the detailed response!
> > >
> > > I'm happy to bump up my score. The results are good (especially when expanded). In principle I would like to see a bit more towards per-layer/per-filter quantization, but there are already some experiments in this direction.
> > >
> > > I do think that making a connection between quantization and pruning is a useful thing for the community to consider too going forward, so I would also argue for acceptance in this direction. This is a nice contribution :-)

---

### Official Review · Reviewer_rnen · 2022-07-11

**Rating:** 5
**Confidence:** 3
**Soundness:** 2 fair
**Presentation:** 2 fair
**Contribution:** 3 good

**Summary:**

This submission proposed to allocate quantization precision for each parameters. It formulated precision allocation as the magnitude of disturbance into parameters: the larger magnitude the parameters can endure, the lower precision should be assigned. During training, a penalty to constraint allocation within budget is added, and the estimated disturbance limit is estimated.

**Questions:**

1. The author mention allocating a precision to each parameter. What is the meaning of assigning precision to each parameter instead of to a layer where parameters within share a same precision ? In my understanding, learn precision in quantization is to fully utilize computational and storage budget: some layers require more precision (quantization bits) to maintain performance while others not. And the quantization's advantage in inference takes effect only when parameters use a same precision in a layer (i.e. in one matrix multiplication). Please explain how the learnt precision is applied to **each** parameter. And if so, how the advantage of quantization is realized.

**Ethics Review Area:**

["I don’t know"]

**Limitations:**

N.A.

**Strengths And Weaknesses:**

Strengths:
1. It is interesting to interpret precision allocation as maximum disturbance in parameters estimation, and the formulation is straightforward.


Weaknesses:
1. The writting is confusing, please refer to the Question part.

---

> ### Author Response · Authors · 2022-08-02
> **Response to Reviewer rnen**
>
> See the above response to all reviewers for a discussion on the applicability of per-parameter precisions and how better real-world efficiency may be achievable with specialized hardware.
>
> In terms of how the quantization operation itself works and how we implement it in the per-parameter precision setting: as an example, a convolutional layer with a weight tensor $w$ of shape $(C_{out}, C_{in}, k, k)$  will be assigned a new set of parameters $s$, which will be a tensor of the same shape as the weights i.e. $(C_{out}, C_{in}, k, k)$.
>
> During the first phase of training (precision allocation), perturbed weights $w + \sigma(s) \odot \epsilon$ are used to compute the layer’s output (i.e. the perturbed weights are convolved with the layer's input), where $\epsilon$ is a tensor with the same shape as $w$ and $s$ whose components are uniformly and independently sampled from $[-1, +1]$, and the addition and multiplication are element-wise.
>
> We then map $s$ to the precisions $p$ with $p = 1 - \log_2(\sigma(s))$ element-wise, with rounding, which yields a tensor $p$ with same shape as $s$: $(C_{out}, C_{in}, k, k)$, and where each component $p_{i,j,k,l}$ will be an integer that represents the precision allocated to the weight component $w_{i,j,k,l}$, for indexes $i,j,k,l$.
>
> To perform quantization we use the standard approach of transforming $w$ to be between $0$ and $2^p-1$, then rounding such that the result is in $\\{0, 1, 2, …, 2^{p-1}\\}$, then doing the inverse transformation to get the final quantized values: the only difference in the per-parameter precision setting is that $p$ is a tensor and the operations are performed element-wise.
>
> For example, if a layer has a 3-dimensional parameter vector $w = (-0.8, 0.1, 0.6)$, we introduce a 3-dimensional $s$ which is trained and then mapped to a 3-dimensional vector $p$, say $p = (1, 3, 2)$. In this case, quantization will yield $Q(w, p) = (-1, 0.25, 0.5)$: each component $w_i$ will be mapped to the closest number that is representable with $p_i$ many bits, e.g. the component $0.1$ of $w$ will be mapped to its closest number in $\\{-1.75, -1.25, -0.75, -0.25, 0.25, 0.75, 1.25, 1.75\\}$ (the set of numbers representable with 3 bits), which is $0.25$.

---

### Official Review · Reviewer_y9nW · 2022-07-23

**Rating:** 6
**Confidence:** 4
**Soundness:** 3 good
**Presentation:** 2 fair
**Contribution:** 3 good

**Summary:**

The manuscript imposes a SMOL framework during training in an attempt to curtail energy consumption of deep learning models and inference time of deployed models by:

- Accounting for task objectives and
- Quantizing parameters into low-precision numeric representations
- Instantiating network weights with full-precision values and auxiliary parameters i.e., learnable noise magnitudes;
- Further, employing above mentioned noise magnitudes as differential proxies for discrete bit precisions which further replace noisy weights (post training)

thus yielding heterogeneous mixed-precision allocation across parameters on ResNet-20 and DCGAN (CIFAR-10), ResNet-18 and ResNet-50 (Imagenet) with benchmarks such as weight quantized CNNs for image classification.

Further exploration of Layer-wise precision allocation and learning structured precision maps are presented in the appendix.

**Questions:**

- The authors are further requested to comment on the **computational efficiency** of the proposed method.
- The claims regarding **higher generation quality** of DCGANs with SMOL has not been algorithmically justified. Why does lower precision representation yield higher generation quality while the loss of the network does not change significantly?
- Supplementary material, Section A.2: Stochastic Approximation, Line 531, what is the **interpretation** of function R?
- Optionally, comment on how the proposed method could potentially **generalize to large language models** (multiplicity of inputs), if at all?
-------------------------------------------------------------------------------------------------
**Additional Note Post-Rebuttal**:
Score has been bumped from _5: Borderline accept_ (Technically solid paper where reasons to accept outweigh reasons to reject, e.g., limited evaluation) to *6: Weak Accept* (Technically solid, moderate-to-high impact paper, with no major concerns with respect to evaluation, resources, reproducibility, ethical considerations).




**Limitations:**

- Inability to represent zero-precision weights has been addressed.
- Computational Complexity is a potential limitation that needs to be addressed (see details above)

**Strengths And Weaknesses:**

**Merits**:
- The paper addresses an interesting and _relatively_ less explored problem i.e., lower-level, mixed-precision numeric representations within machine learning.
- The intuitions regarding precisions and noise perturbations are well-explained!
- The paper additionally accommodates pruning/zero-precision weights.
- Formalization of precision/quantization have been addressed in Appendix.

**Demerits**:
- Claims regarding **energy efficiency, inference time, storage/memory considerations** have not been quantified/mathematically demonstrated/estimated. Provided results (table 2 and table 4) are incremental at best.
- There is no attempt towards **comparisons with standard/widely-used metrics** such as [float16*](https://en.wikipedia.org/wiki/Bfloat16_floating-point_format). The results are not demonstrated on standard networks such as VGG/CNN etc.
*There is a mixed-precision version available as well.

__Minor Corrections:__

- Section 1, Introduction, line 58, the our -> our*
- Section 1, Introduction, line 59, The Appendix provide -> The Appendix provides*
- Section 4, Results, line 187, based on our preeliminary experiments -> based on our preliminary* experiments
- Table 3, Image Generation on CIFAR-10, needs to include the “* represents results with Zero Precision Allocation.”
- In general, highlighting SMOL’s results (through bold font etc.) in tables, helps.
- Supplementary material, Section A.1: Preliminaries, Line 463, the reviewer believes this should be: L($\theta$) = 1/N $\sum_{i=1}^Nl(f_\theta(x_i, y_i))$ where $(x_i, y_i)_{i=1}^N$ is a fixed dataset*.
- Supplementary material, Section A.2: Stochastic Approximation, Point 4, via performs truncation -> via truncation* (?)

---

> ### Author Response · Authors · 2022-08-02
> **Response to Reviewer y9nW**
>
> **Efficiency**: See the above response to all reviewers for a discussion of inference energy costs and some illustrative estimates. In terms of producing models that offer efficient inference, 16-bit representations are not competitive with SMOL, since the average precision allocated by SMOL (and previous methods such as BSQ) is considerably lower.
>
> **Higher generation quality of DCGAN**: Our method has two components which can be seen as some form of capacity control: (1) the noise injected to parameters and (2) the quantization operation – these can have a regularizing effect and result in better performance / generalization as a consequence. For example, in Table 1, SMOL outperforms the full precision model in some settings (93% acc at 2.8bpp compared to 92.6% at full precision).
>
> The performance of networks trained with SMOL is likely a result of some performance degradation due to quantization, along with possible improvements due to the extra regularization caused by the noise injection and the quantization itself (similarly to how dropout at low probability can improve generalization, but harm performance as dropout probability increases).
>
> For the DCGAN experiments, we conjecture that the better FID achieved with 1.7 bpp is due to the additional regularizing effect compared to the 2.5 bpp model. As for the higher performance with SMOL*, we observed that pruning (i.e. allocating zero precision) can result in lower quantization error, since parameters with small value can be set to 0 instead of +1/-1 in the case where they had 1-bit precision; this can lead to smaller performance degradation caused by quantization, even when the average bpp is smaller when comparing SMOL* to SMOL.
>
> **Interpretation of R**: The function $R$ captures the signed quantization error when assigning $p$ bits to a real-valued number $\theta$ and is given by $R(\theta, p) = Q(\theta, p) - \theta$. For example, $R(\theta=1.8, p=2) = Q(\theta=1.8, p=2) - 1.8 = 1.5 - 1.8 = - 0.3$, where we used the fact that quantizing 1.8 with 2 bits yields 1.5. $R$ allows us to represent the quantized value $Q(\theta, p)$ as the original value plus an ‘error’ term, i.e. $Q(\theta, p) = \theta + R(\theta, p)$, where in our framework we use additive noise to estimate how large (in magnitude) $R(\theta, p)$ can be. In other words, $R(\theta, p)$ captures the quantization error and is precisely the quantity we attempt to estimate with the ‘perturbation limits’ described in Section 3.1.
>
> **Language Models**: Our method is architecture-agnostic, in the sense that the procedure of creating and training auxiliary parameters ‘s’ (which designate the magnitude of the noise injected to the original parameters and are then mapped to precisions) can be applied regardless of how the original parameters are used and to what type of layers they belong.
>
> We have run preliminary experiments training a Transformer (6-block encoder and 6-block decoder) on a neural machine translation task (IWSLT'14 German to English) with our method, and we were able to match the performance of the full precision model (34.9 BLEU) with an average of 5.8 bits per parameter. Moreover, we observe that attention layers are assigned significantly fewer bits than the fully-connected layers in the MLP blocks (~3 bpp compared to ~7 bpp), suggesting that attention layers can be efficiently quantized to low precision without degrading the model’s performance – we will add details on these experiments to the paper. We believe SMOL can be used to quantize large language models (BERT, GPTs, Megatron) as well.

---

### Author Response · Authors · 2022-08-02
**Response to all Reviewers**

We thank all reviewers for the valuable feedback and suggestions.

We have collected additional experimental results, including quantizing activations and applying our method to models such as MobileNetV2 and Transformers, which we discuss in the response to Reviewer JFRS – we mention them here in case other reviewers have interest in these experiments.

A common question concerns the practical applicability of models with per-parameter precisions. Though per-layer precisions may be a more natural fit to current GPUs, the hardware and systems research community has been exploring designs which leverage fine-grained and heterogeneous precisions; see [1], [2], and [3]. Our paper focuses on this limit of granularity – where each parameter has its own precision – because it offers the potential to achieve significant energy and efficiency advantages over the per-layer setting. Future commercially available accelerators, based on the latest hardware research, could make these advantages real.

Moreover, while we adopt the per-parameter precision setting for most of our experiments, our method is able to allocate precisions under any level of granularity (per-parameter, per-kernel, per-filter, per-layer) by sharing the newly-introduced variables ‘s’ across groups of parameters. In the case of per-layer precisions, all parameters in each layer share a single scalar ‘s’: this approach is explored in Appendix D.1 and we show that our method outperforms BSQ in this setting. We also ran additional experiments in the per-layer setting on ResNet-20, for example by enabling per-layer pruning, where 2.3 bpp (2.0 bpp with layer pruning) suffices to match the performance of BSQ at 2.7 bpp: this amounts to a 17.4% (35% with layer pruning) increased compression ratio, which is significant. We will add the per-layer experiments to the main paper, as suggested by Reviewer JFRS.

In terms of inference / energy cost estimates for models with per-parameter precisions: the power required to multiply 2-bit and 3-bit weights and a 4-bit activation is 2.41x and 3.83x higher than what is required for a 1-bit weight; while the latency is 1.91x and 2.10x higher, respectively. These numbers are estimated using ripple-carry adder based multiplier designs (which are suitable for low-precision operations) for the corresponding precision settings [4]. For accumulation operations, we assume that the power and latency values are the same for all different precision settings.

Taking the last convolutional layer of a ResNet, which BSQ assigns 2 bits while SMOL assigns 1, 2, and 3 bits to 74%, 24.5%, and 1.5% of the parameters, respectively, we have that the distinct per-parameter precisions allocated by SMOL for that layer require only 57.5% of the power and have 64% of the latency of BSQ’s 2-bit layer. In terms of the energy cost (power x latency), SMOL yields a layer whose energy cost is only 37.3% that of BSQ’s. Note that in real hardware designs, additional control overheads, e.g. around 25% [1], are required to perform fine-grained mixed-precision operations. Although estimates, these calculations suggest that a fine-grained precision allocation scheme can result in significant energy savings on hardware designed to support the corresponding arithmetic operations.

We will add to the paper a discussion of the potential hardware support for fine-grained mixed-precision operations, as well as the associated energy costs and their estimates for models found by SMOL.


[1] E. Park, D. Kim, and S. Yoo. Energy-Efficient Neural Network Accelerator Based on Outlier-Aware Low-Precision Computation. ACM/IEEE Annual International Symposium on Computer Architecture, 2018.

[2] Z. Song, B. Fu, F. Wu, Z. Jiang, L. Jiang, N. Jing, and X. Liang. DRQ: Dynamic Region-based Quantization for Deep Neural Network Acceleration. CM/IEEE Annual International Symposium on Computer Architecture, 2020.

[3] Z. Jiang, Z. Song, X. Liang, and N. Jing. PRArch: Pattern-Based Reconfigurable Architecture for Deep Neural Network Acceleration. IEEE International Conference on High Performance Computing and Communications, 2020.

[4] Stephen Brown and Zvonko Vranesic. 2008. Fundamentals of Digital Logic with VHDL Design with CD-ROM (3rd. ed.). McGraw-Hill, Inc., USA.

---

### Meta-Review · Area_Chair_Q9pf · 2022-08-23

**Recommendation:** Accept
**Confidence:** Certain

**Metareview:**

The paper addresses an important yet less explored problem (optimizing numeric precision per parameters/activations in architectures) with thorough and impressive experiments over multiple architectures (ResNet and Transformers in the rebuttal) and different tasks (ImageNet and machine translation). Reviewers unanimously thinks the paper should be accepted.

**Award:**

No

---

### Decision · Program_Chairs · 2022-09-14

Accept